# Smart textile lighting/display system with multifunctional fibre devices for large scale smart home and IoT applications

Hyung Woo Choi[1,20], Dong-Wook Shin[1,20], Jiajie Yang[1,20], Sanghyo Lee[1,20], Cátia Figueiredo[2], Stefano Sinopoli[3], Kay Ullrich[4], Petar Jovančić[5], Alessio Marrani[6], Roberto Momentè[7], João Gomes[8], Rita Branquinho[2], Umberto Emanuele[3], Hanleem Lee[1], Sang Yun Bang[1], Sung-Min Jung[1], Soo Deok Han[1], Shijie Zhan[1], William Harden-Chaters[1], Yo-Han Suh[1], Xiang-Bing Fan[1], Tae Hoon Lee[1], Mohamed Chowdhury[1], Youngjin Choi[1], Salvatore Nicotera[3], Andrea Torchia[3], Francesc Mañosa Moncunill[5], Virginia Garcia Candel[5], Nelson Durães[8], Kiseok Chang[9], Sunghee Cho[9], Chul-Hong Kim[9], Marcel Lucassen[10], Ahmed Nejim[11], David Jiménez[12], Martijn Springer[13], Young-Woo Lee[14,15], SeungNam Cha[14,16], Jung Inn Sohn[14,17], Rui Igreja[2], Kyungmin Song[18], Pedro Barquinha[2], Rodrigo Martins[2], Gehan A. J. Amaratunga[1], Luigi G. Occhipinti[1✉], Manish Chhowalla[19✉] & Jong Min Kim[1✉]

Smart textiles consist of discrete devices fabricated from—or incorporated onto—fibres. Despite the tremendous progress in smart textiles for lighting/display applications, a large scale approach for a smart display system with integrated multifunctional devices in traditional textile platforms has yet to be demonstrated. Here we report the realisation of a fully operational 46-inch smart textile lighting/display system consisting of RGB fibrous LEDs coupled with multifunctional fibre devices that are capable of wireless power transmission, touch sensing, photodetection, environmental/biosignal monitoring, and energy storage. The smart textile display system exhibits full freedom of form factors, including flexibility, bendability, and rollability as a vivid RGB lighting/grey-level-controlled full colour display apparatus with embedded fibre devices that are configured to provide external stimuli detection. Our systematic design and integration strategies are transformational and provide the foundation for realising highly functional smart lighting/display textiles over large area for revolutionary applications on smart homes and internet of things (IoT).

---

A full list of author affiliations appears at the end of the paper.

**B**reakthroughs in materials[1–6] and process development[7–11] have enabled emerging smart textiles technologies based on electronics with new form factors. Unlike flexible electronics, which have deposited or printed devices on a single flexible substrate, textile electronics does not have limitations on substrate size, dimension of processing tool and mechanical flexibility owing to fibre-structure and continuous weaving process, which encourages the freedom of form factor. The revolutionary system architecture with integrated electronics into fabrics using modern textile engineering principles represents an attractive pathway for realising novel functionalities in textiles[11–16].

Smart or electronic textiles (e-textiles) consist of versatile electronic devices incorporated onto fibre substances[17–22], which have been focused on wearables[23–26]. The recent report on textile system has revealed an example of wearable single coloured textile display with touch sensing and biosignal detection capability[27]. However, a smart textile system over large scale beyond wearable applications has yet to be demonstrated such as curtain lighting/display or digital signage. Moreover, high luminance RGB colour display and various signal detection followed by a real-time illustration on a standalone textile display have not been suggested. The systematic integration of versatile fibre devices into textile over large scale could be realised that meets harsh requirements including (i) material/device design compatible with textile technology, (ii) non-destructive weaving pattern for fibre device, (iii) the interconnection method applicable to textile platform, and iv) instant expression of visual signal from F-device to F-display by signal processing/coding.

Herein, to realise a broad range of multifunctionalities in a single smart textile display system, we integrated one output (fibre LED) and six input devices, which are compatible with symmetric and asymmetric weaving pattern, including; (i) F-radio frequency antenna (F-RF), (ii) F-photodetector, (iii) F-touch sensor, (iv) F-temperature sensor, (v) F-biosensor module, and (vi) F-energy storage that assembled within a natural cotton textile platform. Enhanced control over dimension/performance of devices under mechanical alteration of our F-devices enabled responsive output signal expression after weaving process with long-term stability (over a year). The concept of a smart textile display system for smart homes and real-life IoT applications is built on the developed F-devices delivering processed signals directly to the textile display to enable real-time monitoring/visualising of those signals. F-LED, F-energy storage, and F-temperature were woven at one time while woven F-RF antenna, F-biosensor module, F-photodetector, and F-touch sensor were integrated as Lego-like manner. This Lego-like design is to suggest, (i) post-upgradability, (ii) expanding smart textile system to hundred-inch wide, (iii) seamless operation of textile display with additional textile gadgets. As we have realised our textile system for smart home applications, our prototype of 46-inch system is expected to be much larger when used for real-life applications.

## Results

### Fabrication of smart textile system by weaving process. 
Figure 1 shows the primary steps to realise the fully operational 46-inch smart textile system (34-inch textile lighting/display). We started by imparting specific electronic functionalities onto a single fibre (shape/aspect ratio in Method, Supplementary Note 1), then wove individual fibres into a textile (Fig. 1a). The fabrication method for the smart textile is compatible with standard textile technologies and equipment (manual or machine loom) so that an unlimited size of textile can be fabricated (Fig. 1b). We note that the symmetric weaving pattern was used to build the smart textile, except that the asymmetrical weaving pattern was used to protect an active area of F-devices, especially F-photodetector and F-biosensor module, and to align the F-LED (Supplementary Fig. 1). Then, the novel smart textile system was achieved by the six woven F-devices programmed along with a full-colour 34-inch display (F-LED) (Fig. 1c). All fibre devices were connected using highly conductive fibres (developed in-house, Ag-coated polyamide (Ag-PA), Supplementary Fig. 2, Tables 1 and 2) that allow signal transfer from an F-device to an assigned controller that relays the information to the F-LED display (output). The nature of the textile enables the entire system to be folded, rolled, bent for wall-mountable curtain lighting/display (less than 5 mm-thick, Fig. 1d). We first interpret the performance of F-devices comprising the smart textile system along with mechanical and electrical stability followed by explaining the operation principle of the system.

### Fibre devices characteristics. 
All F-devices fitted to weaving/interconnection process are uniquely designed and optimised for our textile system. Each F-device involved in the smart textile system shows core-shell structure or is fabricated onto a single fibre. As a pivotal output device (F-LED), lighting and display apparatus, the textile lighting/display consists of $84 \times 76 \times$ RGB LEDs ($1.91 \times 10^4$ subpixels) mounted onto copper fibres and woven with cotton fibres line by line asymmetrically (ratio 1:3 for F-LED versus cotton thread) to avoid distortion of visualising images (Supplementary Fig. 3). We also developed the F-LED with $120 \times 65 \times$ RGB LEDs ($2.34 \times 10^4$ subpixels) to enhance the resolution. It is noteworthy that no higher resolution of textile display with fibre LEDs has yet been demonstrated. Further reduction of the inter-LED distance led to the interference between neighbouring pixels resulting in inhibiting single-pixel control. At low operation voltage (<3 V), the luminance exceeded $10^4$ cd/m² for RGB colours while maintaining their colour purity and brightness under shape change (Supplementary Fig. 3e, Movie 1, 2).

F-RF antenna, a square spiral antenna designed to receive the electromagnetic field at the distances from the RF source (frequency of 13.56 MHz) (Fig. 2a) was fabricated by embroidery method using Ag-PA conductive fibres (20 filaments/thread, Supplementary Fig. 4) for a mode switch ('display mode' to 'monitoring mode'). The antenna properties and design rules are summarised in Supplementary Fig. 5a,b and Table 3. The inductance of antenna is proportional to the number of turns. In contrast, the width of an embroidered straight line is inversely proportional to the inductance[28–30]. Total inductance of embroidered antenna was $L_{measure} = 2.64 \mu H$ (Supplementary Fig. 5c, the deviation is 6.04%, compared to the simulated inductance, $L_{Wheeler}$). The impedance of F-RF was $95.3 + j220$ $\Omega$ at 13.56 MHz (Supplementary Fig. 5d).

F-photodetectors consist of aluminum (Al)/zinc oxide (ZnO)-graphene photoactive layer/epoxy encapsulation layer structure on polyethylene naphthalate (PEN) fibres. Eight-channel F-photodetectors connected in parallel in the textile woven with the asymmetrical pattern were tested via UV irradiation (0.5 mW) (Fig. 2b). The results show that a detector generates a maximum of ~100 times increase in photocurrent at $V_{bias} = 10V$ and exhibits average rise and decay times of 3 sec and 1 sec, respectively (Supplementary Fig. 6). Compared to the state-of-the-art flexible ZnO-based photodetectors, our F-photodetector that was integrated into the textile under harsh mechanical stress relatively exhibits low operating voltage and fast rise (decay) time along with comparable on/off ratio (Supplementary Fig. 6).

Network of F-touch sensors is achieved using Ag-PA conductive fibres that exhibit a change in resistance (converted to the output voltage signal as the function of touch duration

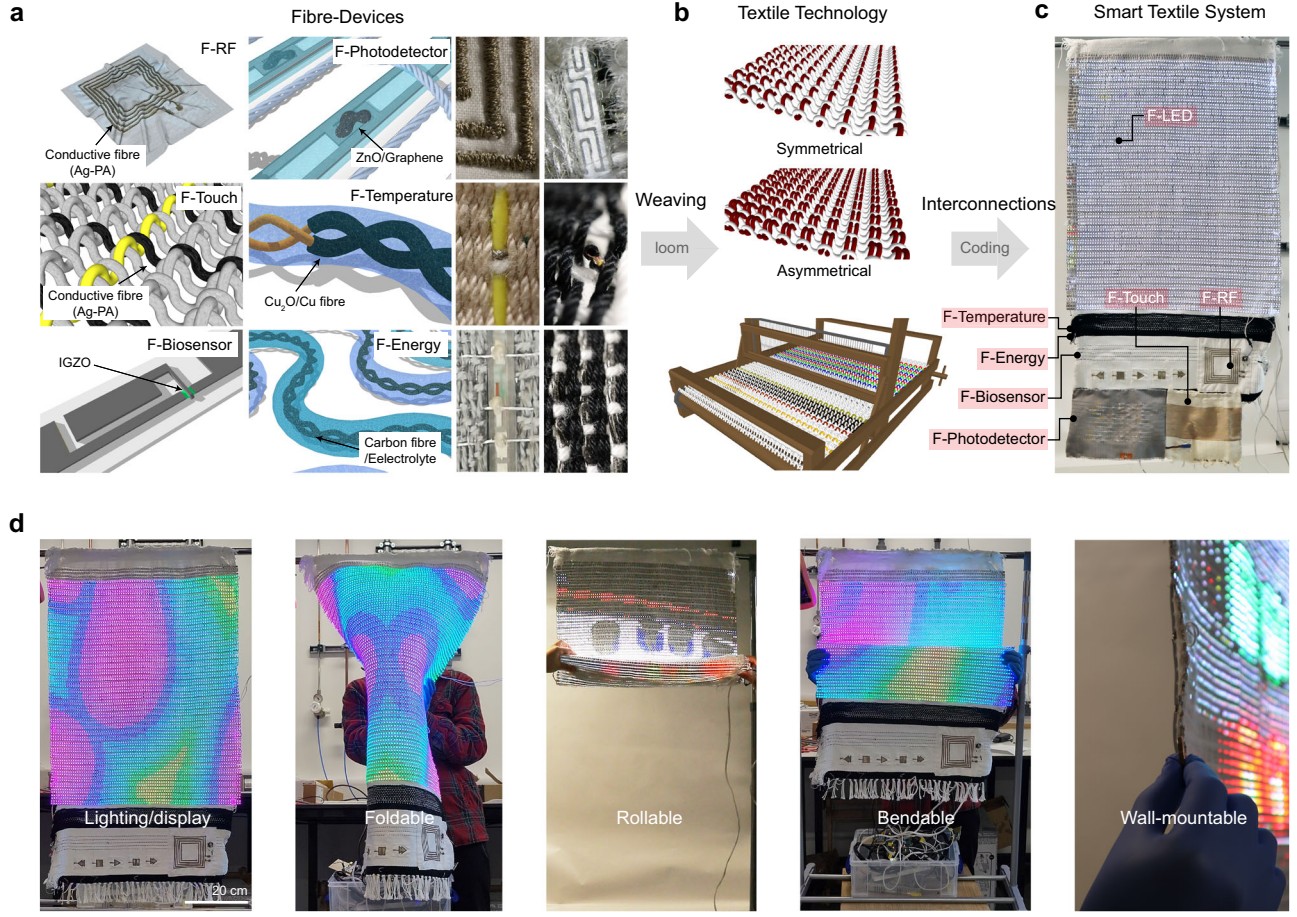

**Fig. 1 The fabrication protocol and mechanical stability of smart textile system integrated with six F-devices and lighting/display apparatus (F-LED).** Technology convergence between textile engineering and electronic science for smart textile display system fabrication; **a** Schematics of F-devices and corresponding fabricated F-devices. **b** Textile technology (weaving). **c** Smart textile system. **d** Mechanical stability of smart textile display system under folding, bending, rolling conditions and side view of the wall-mountable system.

from 1 s to 30 s) when touched (Fig. 2c, Supplementary Fig. 7). Those conducting fibres were woven into the textile in the weft (horizontal) and warp (vertical) directions with spacings of 2 and 3 cm, respectively. The weft conducting fibres are encapsulated by a polyolefin tube (yellow in Supplementary Fig. 7) while vertical fibres are left exposed to air. This method allows complex, simultaneous, and multi-touch-read functions that are comparable to a touchpad. An electrical readout circuit acquires the change in resistance ($\triangle R(R_{touch} - R_{release})/R_{release} \times 100 \geq 2\%$) as fast as 1 Hz. F-touch sensor shows the response times of 18 ms from noise to the maximum resistance change (Supplementary Fig. 7). Signal-to-noise ratio ($SNR = 20\log_{10}(V_{signal}/V_{noise})$) exhibits higher than 70 dB, which is sufficient for identifying active/inactive status.

Two core-shell copper/copper oxide (Cu/Cu$_2$O) fibres twisted together were used to realise F-temperature sensors woven in the textile system, which show that the resistance continuously decreases as a function of increasing temperature from 5 °C up to 70 °C when placed on a hot plate (Fig. 2d). Cu$_2$O is a well-known negative temperature coefficient material (Supplementary Fig. 8)[31,32]. The characteristics of F-temperature sensor fit well with the Steinhart-Hart model[33], as shown in Fig. 2d. In order to evaluate the sensitivity of F-temperature sensor, temperature coefficient of resistance of 1.44 ± 0.36 %/K at the reference temperature of 25 °C was extracted (Supplementary Fig. 9a, b), which shows an estimated resolution of approximately 0.1 °C

(Supplementary Fig. 9c) similar to state-of-the-art F-temperature sensors[2,34]. The most sensitive temperature range for the F-temperature sensor is found to be between 5 °C to 70 °C that is applicable for real-life indoor/outdoor conditions. The origin of thermal conduction can be found by the extracted thermal activation energy of Cu$_2$O that is 0.266 eV, which is consistent with copper vacancy level ($V_{Cu}$) from valence band maximum (Supplementary Fig. 9d)[31].

F-biosensor module based on an amplifier in a common-source configuration consists of F-transistor and a resistor along with an electrocardiogram (ECG) pad (Fig. 2e, Supplementary Fig. 10a). The woven transistor was achieved by the asymmetrical weaving pattern along with laser soldering with Ag glue within 1 sec (details of laser soldering and alignment in Supplementary Fig. 10b–d and material property in Table 4). Tailored fibre transistors with indium gallium zinc oxide (IGZO) as the channel exhibit threshold voltage ($V_{TH}$) = 0.9 V, on/off ratio = $10^7$, saturation mobility ($\mu_{sat}$) = 8 cm$^2$/V·s that shows one-year stability in ambient environment (Supplementary Fig. 11a,b). Theoretical amplification gain (defined as $A_v = g_m R_D$)[35] can be obtained from transconductance and resistance of a load resistor ($R_D$). To maximise empirical voltage gain ($A_v = \partial V_{out}/\partial V_{in}$), a resistor of 70 MΩ was connected as a bias load to ensure that the heartbeat signal (typically less than 1 mV) reaches a voltage gain of ~28 V/V at its peak (Fig. 2e, Supplementary Fig. 11c). As the device operated at the bias current of 131 nA in the saturation

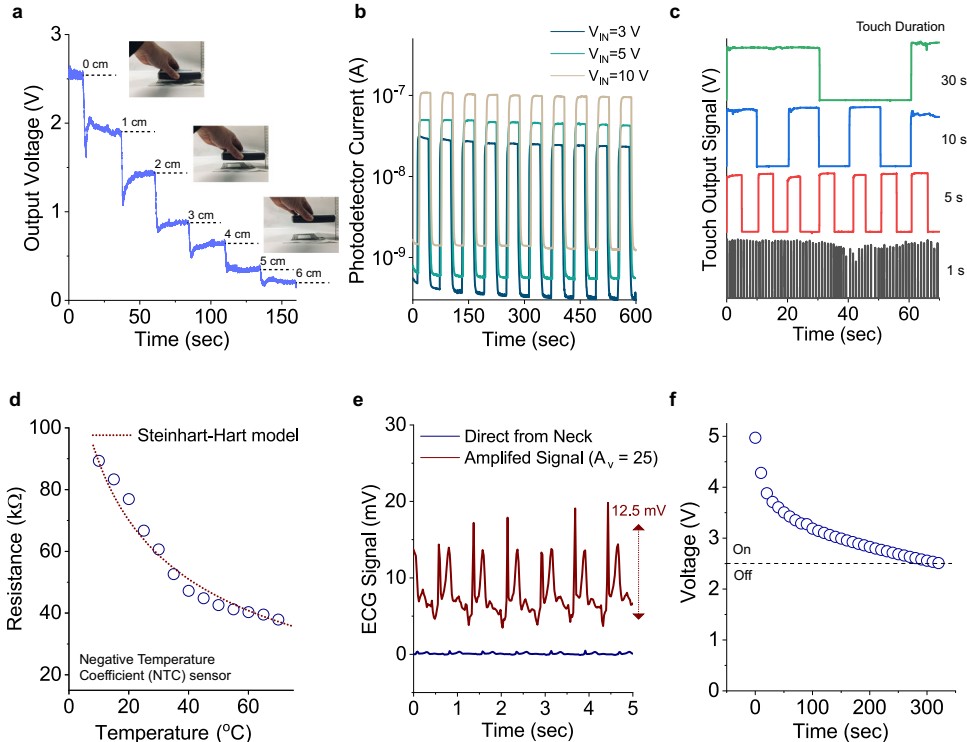

**Fig. 2 Characteristic of six F-devices. a** F-RF antenna, Output voltage generated from F-RF antenna with a suppressing resistor demonstrates distance-dependent signal classification. **b** $I_{light}/I_{dark}$ as a function of time of F-photodetector under various input voltage under UV on and off. **c** Touch output signal as a function of touch duration. **d** Negative resistance changes as a function of temperature of $Cu_2O$ F-temperature sensor. **e** Amplified (gain ~ 25) heartbeat by F-biosensor module. **f** Serially connected fibre supercapacitors as power switch of textile lighting/display (on/off).

regime, the peak power consumption was found to be 350 nW indicating low power consumption as a peripheral module in the smart textile system (Supplementary Fig. 11d).

Double twisted fibre supercapacitors (F-energy storage) consisting of gel-electrolyte (polyvinyl alcohol (PVA)/$H_3PO_4$) between carbon fibre (CF) bundles have also been integrated into smart textile system as a trigger switch between power supply and textile lighting/display instead of the main power source as the power consumption of output lighting/display LED is measured to be a maximum of 35 W (Fig. 2f, Supplementary Fig. 12). Five serially connected fibre supercapacitors show the total output voltage of 5 V (discharging curve during operation of textile lighting/display, display-off below 2.6 ± 0.1 V, Fig. 2f). To increase the capacitance of fibre supercapacitor, conductive fibre (Ag-PA) was incorporated into CF bundle resulting in a 3-fold increase in total capacitance (Supplementary Fig. 12). Fibre supercapacitors exhibit an average capacitance of 5.47 ± 0.78 mF (CF only) and 17.5 ± 0.6 mF (CF with Ag-PA) at a scan rate of 100 mV/s. Among fibre supercapacitors (electrical double-layer capacitors) reported to date, our supercapacitor with conductive fibre shows high specific energy and power output.

**Mechanical and electrical stability of fibre devices.** Strong emphasis on mechanical and electrical stabilities has been given to fibre devices[14,36]. To evaluate the mechanical stability and electrical reliability of each F-device, cyclic bending under 10 mm radius or stretching tests have been performed. F-LED, as the output device, maintains initial electrical characteristics over a year (Supplementary Fig. 13) and shows unnoticeable current density change under 1000 bending cycles ($\triangle V_{J=100mA/cm^2} = 0.2V$, $\triangle J_{1000cycle,r=10mm} = \pm 5mA/cm^2$) owing to highly flexible copper fibre (Fig. 3a).

In the case of conductive fibre, we note that it does not alter its conductivity (approximately $2.32 \times 10^6$ S/m, Fig. 3b) against

10 mm bending condition so that the harsh stretching test was performed (Supplementary Fig. 14). A conductive fibre that is crucial for transferring electronic charges from F-devices to controllers, as well as the essential material for F-RF antenna, F-touch sensor, F-energy storge, and device interconnection, was located on step-motored elongation stage. Continuous cycling strain up to 20% showed constant current of $1 \times 10^{-4}$ A under $1 \times 10^{-3}$ V. It was found that uni-axial strain of more than 35% resulted in a breakdown of conductive fibre that is significantly overqualified for mechanical bending or rolling condition of the entire smart textile (Supplementary Fig. 15)

F-photodetector, fixed bending radius 10 mm, was tested under cyclic mechanical manipulation (Fig. 3c). At the bias voltage of 10 V, off-current and on-current were continuously maintaining near 9.3 nA (relative standard deviation: 5.82%) and 278 nA (relative standard deviation: 8.62%), respectively that confirmed the electrical stability during 1000 bending cycles. It is clear that 'on-off' current still lies in the distinctive range that is substantial enough to categorise photonic input identification.

F-touch sensor is based on the resistive mode responding to finger contact. Weariness of conductive surface layer would result in a non-resistance change at a contact position. After operational touch intensity (on-status) was set to $\triangle R/R_{max} = 0.5$ (Fig. 3d), 1000 touch attempts (3000 touch results in Supplementary Fig. 7h) were performed leading to an average of 0.78 ($\triangle R_{avg}/R_{max}$, relative standard deviation: 4.86%) that is indicative all the touch signals displayed higher than 'on-status', in turn, the conductive layer (Ag) was not affected during repeating touches.

F-RF antenna is made of conductive fibres, which shows slight resistance change during 1000 bending cycle (Fig. 3e). As comparable with single conductive fibre characteristics in Fig. 3b, the electrical current along with spiral square antenna was maintained that indicated the unchanged antenna property in

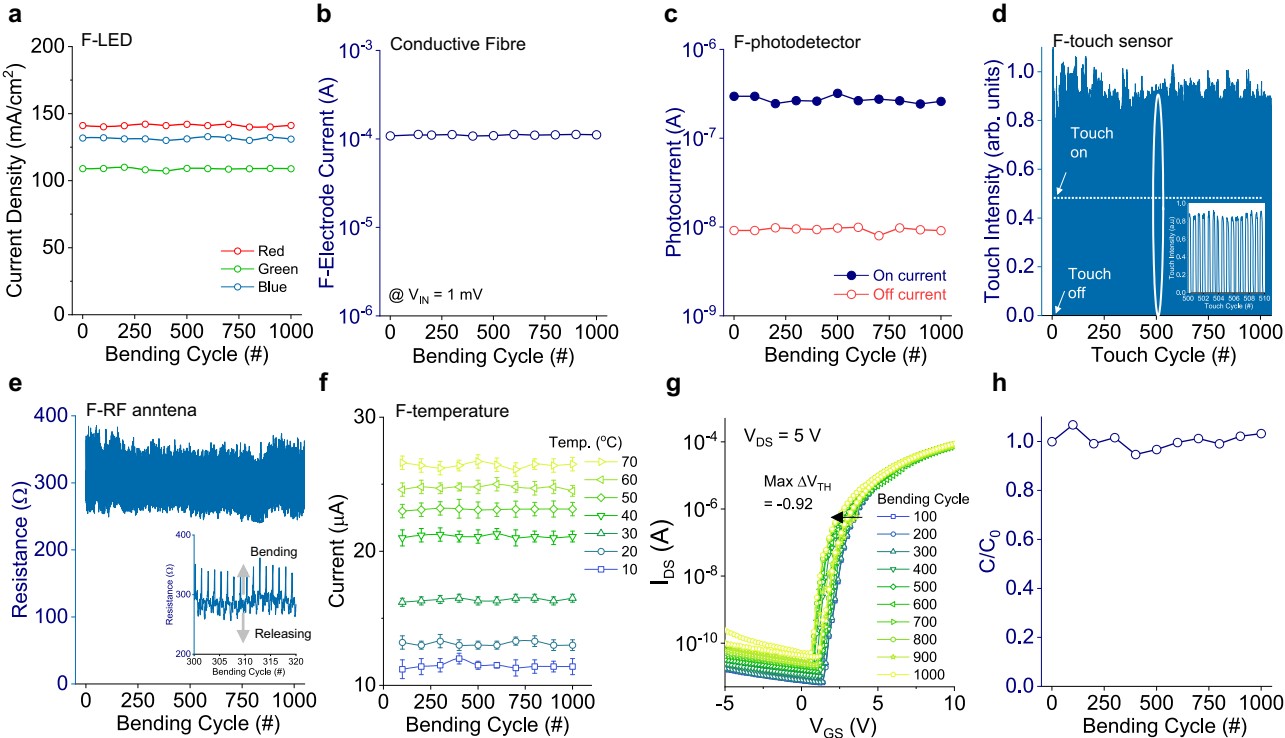

**Fig. 3 Mechanical and electrical stability of F-devices, 1000 cyclic bending test with 10 mm radius. a** F-LED showing no current density change. **b** Stable current of conductive fibre. **c** F-photodetector showing clear on/off classification. **d** Cyclic touch attempts for evaluating surface weariness of F-touch sensor. **e** Stable resistance value of RF-antenna (Ag-PA conductive fibre). **f** F-temperature sensor showing stable classification of measured temperature range as a function of electrical current. The error bars on the plot represent the standard deviation of measurements. **g** Unnoticeable transfer characteristics change of fibre transistor used for F-biosensor module. **h** Mechanical damage-free from the bending for fibre supercapacitor.

term of output voltage, output power, rectified signal against the distance (Supplementary Fig. 16).

F-temperature sensor sustained its original conductivity under bending conditions of 10 mm radius (Fig. 3f) because the active layer mechanically protected by the encapsulation layer. However, owing to the oxide material nature, even F-temperature sensor was encapsulated by epoxy, the resistance was increased over one-year period (e.g. at T = 40 °C, $R_{originanl}$ = 47 kΩ, $R_{1-year}$ = 263 kΩ, Supplementary Fig. 9), because of further oxygen diffusion into F-temperature sensor originated from water (moisture) vapour transmission rate of epoxy in ambient[37]. We note that the increasing current trend as a function of temperature was maintained for segmenting temperature levels, which can be compensated by adjusting programming code variables.

When fibre transistor of F-biosensor module is subjected to bending, the electrical characteristics are influenced due to channel and dielectric leakage current under mechanical stress. To identify the working condition limit, 1000 cycle bending with radius of 10 mm (Fig. 3g, lower bending conditions in Supplementary Fig. 11e) was applied resulting in unnoticeable transfer characteristics change ($\triangle V_{TH}$ = −0.92 V, on/off = $10^7$) that is suitable for F-biosensor module. The n-type channel material (IGZO) with encapsulation layer (parylene-C) does not show a notable change in its mobility ($\triangle\mu_{sat}$ < 0.45 cm²/Vs) and threshold voltage ($\triangle V_{TH}$ < 0.5 V) over a year (Supplementary Fig. 11a, b) that reflects material stability is achieved.

Bundle of CFs with or without conductive fibres in F-energy storage device is damage-free from 1000 bending cycles (radius of 10 mm) confirming mechanical manipulation of smart textile is viable (Fig. 3h). Besides, the electrolyte (PVA-H₃PO₄) layer does not show any noticeable capacitance degradation over a year

($\triangle C$ = 0.26 mF) or during 1000 cyclic charge-discharge tests (over 0.95), which further reveals electrochemical stability of F-energy storage by ideal capacitance retention (Supplementary Fig. 12).

Further investigation of mechanical reliability of F-device-embedded textile is performed by abrasion test for three functional textiles which are subjected to be touched, pressed or possible fatigue over a time without protective layer (Supplementary Fig. 17); (i) LED-embedded cotton textile, (ii) conductive fibre embroidered cotton textile, and (iii) conductive fibre embroidered polyester textile. The standard disc-type Martindale-abrasion test revealed that RGB LED and conductive fibres maintain their original properties (luminance of 1000±10 cd/m² and conductivity of $2.32 \times 10^6 \pm 0.5$ S/m) after 1000 cycles of abrasion test under 1 kg load which is suitable for decorative use in non-wearable applications (ISO 12947-1).

**Operation principle of smart textile display system.** As a proof-of-concept of smart textile lighting/display system, we connected six devices including F-RF antenna, F-photodetector, F-temperature, F-touch sensor, F-biosensor module to textile display, and F-energy to display power switch and utilised the integrated textile system as a smart home appliance. The working principle of the integrated smart textile system is summarised in Fig. 4a; incoming signals from individual F-devices (by electromagnetic, photonic, physical, environmental, and biological) require several steps of signal process and categorisation, then being visualised on the textile display. Several controllers are connected to F-devices with a specific circuitry to operate the smart textile display system (Photo in Fig. 4a).

The textile display in the smart textile system can work in both 'display mode' (grey-level-controlled moving pictures and lighting as shown in Supplementary Movie 1 and 2) and 'monitoring

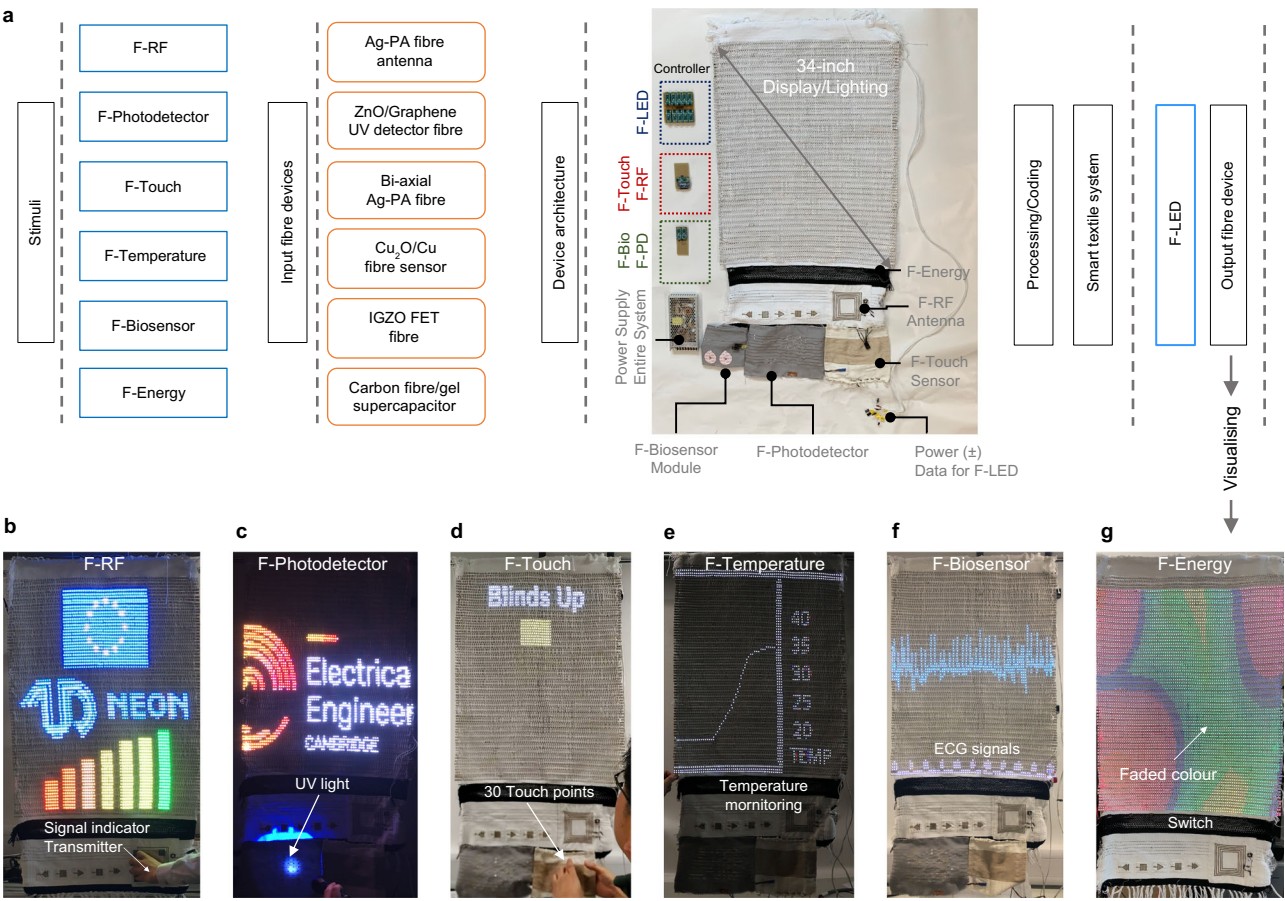

**Fig. 4 Design of smart textile display system from material to system level. a** List of F-devices integrated into the smart textile system, materials, device architecture, auxiliary parts, signal processing/coding, and F-LED visualising. Applications displaying on textile system: Real-time operation photos of **b** Six-signal strengths digitized depending on the distance between F-RF antenna and transmitter. **c** Programmed image initiated by UV irradiation on F-photodetector. **d** Real-time temperature monitoring by F-temperature sensor grasped by fingers. **e** Electrocardiography signal measured by F-biosensor module. **f** One of 30 IoT functions operated by F-touch sensor. **g** F-energy as a power supply to the switch of textile display.

mode' (still images presenting the information from each F-device, Supplementary Movie 3 to 7, examples of RF signal strength, photodetection, touch, temperature, and biosignal) using F-energy storage as a switch between main power supply and textile display (Supplementary Movie 8).

F-RF, F-photodetector, F-temperature, and F-biosensor module activated by radio frequency waves (13.56 MHz), UV light (365 nm), ambient temperature, and biosignal, respectively, as contactless devices inform electromagnetic field contents, weather elements, and cardiac activity.

As the RF-transmitter is brought near an embroidered F-RF square spiral antenna, the response, *e.g.* a colour-bar chart (classified at 2.5, 2, 1.5, 1, 0.7, 0.4, 0.2 V, Fig. 4b, Supplementary Fig. 16 and Movie 3), appear on the textile display depending on a signal reception range (0 to 6 cm). At the maximum detection range (6 cm), the textile display turns from 'display mode' to 'monitoring mode' as rectified input signal triggers mode change, whereas the withdrawal of RF-transmitter away from the antenna instantly draw the mode back to 'display mode' which indicates the removal of RF signal.

The motivation for F-photodetector and F-temperature integration in the smart textile system stems from environmental monitoring. UV detection function of F-photodetector is verified by that the smart textile system displays the university deparment logo upon UV illumination on light-sensitive layer of ZnO-Graphene (Fig. 4c, Supplementary Fig. 18 and Movie 4). When the system is installed on the window, *e.g.* smart curtain,

F-photodetector facing outward reads UV intensity. Dependence on UV protection range from 15% up to 100% (under AM1.5 G) shows F-photodetector current ranges from $5 \times 10^{-8}$ A to $3 \times 10^{-7}$ A that allows determination of window coverage (Supplementary Fig. 19, roll-up/down smart textile curtain). F-temperature sensor was introduced to monitor indoor/outdoor environment temperature. It was used to obtain the resistance values from which a real-time temperature was calculated and displayed (Fig. 4d, Supplementary Fig. 20). To monitor temperature response, grasping F-temperature sensor by fingers was performed to mimic fast environmental temperature change that revealed spontaneous temperature response (Supplementary Movie 5).

To read the heartbeat from one of the authors (Fig. 4e and Supplementary Movie 6), the output voltage from F-biosensor module was used. F-biosensor module is able to amplify 0.5 mV from the body signal to 12.5 mV (empirical voltage gain $A_v = \partial V_{out}/\partial V_{in} = 25 \pm 2$) and the real-time heartbeat is plotted on the textile display without the use of an external amplifier (Supplementary Fig. 21, bpm = 80, 120 before/after running, respectively).

F-touch device is activated by physical stimuli (contact mode). F-touch sensors were designed to operate each fibre device as well as show a set of pre-coded instructions on the textile display when touched (Fig. 4f, Supplementary Movie 7). The circuitry was made by interconnecting Ag-PA fibres with touch sensors array (Supplementary Fig. 22). To implement practical real-life

applications, thirty execution terms for smart gadgets and internet of things (IoT) were demonstrated to every sensor element (Supplementary Fig. 23 and Table 5).

F-energy storage was connected between textile display and the main power supply (35 W, $V_{output}$ = 5 V) to turn on the textile display by applying a voltage to the switch (Fig. 4g, the configuration in Supplementary Fig. 12a, Supplementary Movie 8). After the operation for 300 sec resulting in voltage drop to 2.6 ± 0.1 V, the textile display turns off.

Thermal stability of textile display in the smart textile system was tested as a safety measure and showed that the temperature variation of less than 2 °C (from 24.7 to 26.2 °C) was observed over a continuous operation for 6 h (Supplementary Fig. 24). The standard water resistance property of all fibre devices in our textile system has been investigated under IPX7 condition (1 m deep, 30 min) and revealed that all F-devices exhibited unnoticeable performance change (Supplementary Movie 9, Supplementary Fig. 25).

We have successfully merged textile engineering and fibre-based electronics in order to integrate a fully functional versatile smart textile system. A large (46-inch) smart textile system exhibits high luminance RGBW lighting/display coupled with six functional fibre devices capable of real-time electromagnetic, physical, signal monitoring with freedom of form factor. Our systematic integration of multifunctional F-device into textile platform will be a cornerstone and give a guideline for not only electronic engineers but also traditional textile engineers in the emerging smart textile field to overcome dimensional and form factor limitations.

## Methods

**Weaving process.** To weave the smart textile system with six F-devices and F-LED (width: 60 cm), the number of weft-warp threads counts more than a thousand. Precise design of entire textile system and position of electronic F-devices should be considered prior to weaving process. Mechanical and performance degradation during the weaving to F-RF, F-touch sensor, F-temperature sensor, and F-energy storage is negligible owing to conductive fibre-based device architecture and a protective coating layer. In contrast, F-photodetector and F-biosensor module are weak to mechanical abrasion during the weaving. To increase openness of active area of those mechanically sensitive F-devices, we adopted an asymmetrical weaving pattern that those unwanted fibres detour to the peripheral position of F-devices. An asymmetrical weaving pattern is also required to prevent electrical short-circuit between conductive fibres which are in contact with electrodes of F-devices (Supplementary Fig. 1). Considering these critical pattern-related parameters, the number of conductive threads in warp direction should be determined before the weaving process to isolate the functional device without electrical interference. At the same time, the number of non-conductive threads in both warp/weft directions should also be considered before the weaving process as the density of threads influences the maximum bending radius owing to interlacing tension.

**Fabrication of conductive fibre.** A feeding reel has 1-kilometre-long pre-treated polyamide (PA) fibre (Supplementary Fig. 2). Pre-treatment, either surface roughening or seed-layer (platinum, Pt) deposition, was done for promoting silver nanoparticle adhesion and nucleation/growth. The raw fibre feeding speed (Supplementary Table 1) and minimum/maximum of deposition speed, temperature, volume of electrolyte, current density, Ag layer thickness during the electroplating step were adjusted depending on Ag thickness and relative conductivity (Supplementary Table 2). After electroplating of conductive layer, two steps of washing were done to remove residual electrolyte. Drying of Ag-PA fibres was done in a vacuum box before collecting fibres on retrieving reel.

**Fabrication of fibre LED.** Full-colour RGB F-LED that allows the dynamic visual monitoring of input stimuli detected by F-devices were made by modified conventional LED pick-place method[38,39]. The textile display in the smart textile system consists of display size of 34-inch and a screen resolution of 84 × 76 and 120 × 65 pixels. Designed pixel-to-pixel pitches on the F-LED were 7 mm and 5 mm, respectively. To achieve a better resolution textile display, we re-designed a flexible copper fibre (width 4 mm) with patterned electrodes by chemical copper etching. Solder paste was applied to each LED mounting location. After placing LEDs aligned with soldering paste on a copper fibre, hot-dry reflowing (< 250 °C, solidifying solders) was performed. To achieve single-pixel operation, overlapping

of reflowed-solder paste was inhibited by leaving the least amount of solder paste (Supplementary Fig. 3a).

Interconnection between following F-LED was done by conductive fibre (Supplementary Fig. 3b). Two power lines (±) with one data line were connected at the one end of each F-LED. Power and data lines were connected by zig-zag route to define the positions of LEDs. To minimise image distortion, the horizontal position of F-LED was adjusted during the weaving process (Supplementary Fig. 3c, d). A controller has limited data accessibility up to 700 pixels, which indicates we added one controller every eight LED fibres, leading to 10 controllers that were required to operate a 34-inch textile display (photo of all controllers in Fig. 4a). In J-V-L curves of RGB LED, the luminance consistently increased with increasing operating voltage, transcending the industry-standard luminance (Samsung, LG, and Signify) for lighting (> 10,000 cd/m²) and large display (>1,000 cd/m²) purposes at low operation voltage below 3 V (Supplementary Fig. 3e).

**Fabrication and design rules of F-RF antenna.** We investigated a textile RF antenna with a square spiral shape that has the advantage of operating over a wide range of frequencies by length adjustment and increasing inductance by adding multiple turns of the pattern. The dimension of the F-RF antenna is first determined by width (W) and length (L) of 2-dimensional shape (2D). Our antenna was designed to be a square spiral type (W = L). Ag-PA conductive fibres were used to form a square spiral antenna by embroidery method on cotton fabric (Area$_{cotton}$ = 400 cm², Supplementary Fig. 4). To achieve higher conductivity, twisted threads (20 filaments per thread) were fed to the embroidering machine by transferring a custom-design pattern. To convert RF signal (AC) to DC voltage, a rectifier bridge was made by connecting 4 diodes (RS1G E3 400 V SMD diode) where two-terminals from the antenna connections were marked V1, V2 as incoming RF signal (Supplementary Fig. 4).

Calculated antenna properties and design rules of the square spiral antenna are summarised in Supplementary Fig. 5 and Table 3. Based on a modified Wheeler formula and expression on current sheet approximation[28,29], the inductance of antenna (shape, length (L), width (W), line width (w), line spacing (s), and connectors) was calculated to maximise the signal reception as shown in Supplementary Fig. 5. When a squared antenna design is considered, based on the modified Wheeler formula, the inductance of the antenna is given by (parameters are summarised in Supplementary Table 1)[28,29,40,41];

$$L_{wheeler} \approx K_1 \cdot \mu_0 \cdot N^2 \cdot \frac{D_{avg}}{1 + K_2 \cdot \delta} \quad (1)$$

The overall inductance of an antenna is determined by parameters including; the average diameter of the antenna, $D_{avg} = 0.5(D_{IN} + D_{OUT})$, $K_1$ and $K_2$ are Wheeler parameter indexes (inductance reduction factors), $\mu_0$ is the permeability of free space ($4\pi \times 10^{-7}$ H/m), N denotes the number of turns, and $\delta$ is filling ratio ($D_{OUT} - D_{IN}/D_{OUT} + D_{IN}$). Another simple and accurate expression including the antenna's layout parameters is also employed based on approximation of one side of a spiral antenna as a current sheet approximation (CSA model).

$$L_{CSA} \approx \frac{1}{2} \cdot \mu_0 \cdot N^2 \cdot D_{avg} \cdot c_1 \cdot \left( \ln\left(\frac{c_2}{\delta}\right) + \frac{c_3}{\delta} + c_4\delta^2 \right) \quad (2)$$

where, $c_1$, $c_2$, $c_3$, and $c_4$ are dimensional correction parameters (Supplementary Table 3). When both equations were used for the calculation, their difference in the total inductance does not heavily rely on the calculation models (Supplementary Fig. 5a). In contrast, the number of turns, the width of conductive lines, and outer length (or width) ($D_{OUT}$) of antenna have more influence over other parameters (Supplementary Fig. 5b, c). Discrepancies of values obtained from measurements and calculations ($discrepancy$ (%) = $100 \times (L_{meas} - L_{wheeler})/L_{wheeler}$) stem from embroidered nature of the F-RF antenna that omits void-free, two-dimensional planar conductive lines on an insulating substrate (cotton or polymeric substrate).

**Fabrication of F-photodetector.** ZnO nanoparticles were synthesized according to our previous report with minor changes[42,43]. Zinc acetate dehydrate was dissolved in methanol (4.5 mmol) in a two-neck flask equipped with glass condenser, followed by heating up to 60 °C in air. Potassium hydroxide (0.469 g) was dissolved in methanol (22 ml) and added to zinc acetate solution by a syringe pump for 10 min. Mixed solution was kept at 60 °C for 90 min. After cooling down to room temperature, ZnO nanoparticles were collected by centrifuging at 7025 × g for 10 min, then redispersed in 30 ml of methanol. This purification process was carried out at least twice to remove the unreacted materials. The graphene flakes were prepared by the liquid-phase exfoliation method, according to our report[44]. 10 mg/ml graphite flakes in N-Methyl-2-Pyrrolidone were ultrasonicated for 9 hours followed by washing in ethanol three times. To inhibit aggregation of exfoliated graphene flakes, 0.1 ml ammonium solution to 100 ml graphene solution was added followed by washing/re-dispersion in ethanol three times. The mixed solution of ZnO and graphene with the addition of ethylene glycol and isopropanol was prepared for drop-coating. Before coating of ZnO/graphene film, aluminum electrodes on PEN fibre were deposited by thermal evaporation with a shadow mask. The drop-coated ZnO/graphene film was post-annealed at 150 °C for 10 min to remove residual

solvent (Supplementary Fig. 6). To protect the device during the weaving process, optical adhesive (Norland 65) was deposited only on top of photoactive layer.

**Fabrication of F-touch sensor.** The first step of weaving process is to align warp (vertical direction) threads on a loom. As our F-touch sensor features $5 \times 6$ (horizontal × vertical, total 30 sensor points) arrangement, the distance of conductive fibres (Ag-PA) was determined to be 8-threads counts in vertical direction (3 cm, Supplementary Fig. 7). The distance of horizontal conductive fibres was maintained for 2 cm to avoid touch interference.

**Fabrication of F-temperature sensor.** Copper fibres ($\Phi = 500\ \mu m$) were annealed in air to oxidize surface, resulting in the formation of $Cu_2O$ layer ($<10\ \mu m$, Supplementary Fig. 8a) on the surface of copper fibre (core/shell structure). Combustion was initiated by using a butane ($C_4H_{10}$) torch in the air for 1 min with a consequence of surface oxidation ($T_{annealing} < 287\ °C$). X-ray diffraction (XRD) pattern confirmed $Cu_2O$ phase (Supplementary Fig. 8b,c). Two core-shell type Cu/$Cu_2O$ fibres (Supplementary Fig. 8d) were twisted to meet the requirement for temperature sensor while measuring resistance until the resistance value lied between $10\ k\Omega$ to $1\ M\Omega$. Overlapped part of twisted Cu/$Cu_2O$ (length: 5 mm) was encapsulated by epoxy to prevent further environmental effects (moisture, oxygen, etc.). Copper fibres (around 60 cm) left over after the twisted Cu/$Cu_2O$ were woven into the textile system, then utilised to connect into a signal analyser by conductive fibres.

**Fabrication of F-biosensor module and detail interconnection strategy.** A fibre transistor was made based on our previous report with slight modification[45,46]. Gate electrode, Molybdenum (Mo) was deposited by a sputter in pure Ar atmosphere on PEN (125 μm thick) substrate. The thickness of Mo was measured to be 60 nm. Dielectric layer of 160 nm-thick, seven-layer stacks composed of alternating $SiO_2$–$Ta_2O_5$ (1 nm/min, 3.4 nm/min deposition rate, respectively)[47] is sputtered in $Ar/O_2$ to obtain high capacitance 96 nF/$cm^2$ with mechanical flexibility. A 40-nm thick indium gallium zinc oxide (IGZO) semiconductor was deposited by sputtering (2:1:1 atomic ratio of In:Ga:Zn). As a top contact configuration, 60-nm thick Mo was sputtered for source and drain electrodes. The semiconductor layer was patterned by the lift-off process while the gate electrode was patterned by reactive ion etching by sulfur hexafluoride ($SF_6$) (Supplementary Fig. 10a). To protect entire transistor area, parylene-C layer (1.2 μm) was deposited by chemical vapour deposition followed by etching all electrode (gate, drain, and source) parts for conductive fibre connection. A flexible substrate was cut into fibre shape (width = 3 mm) by a laser cutting tool. The interconnection from fibre transistor to an external power source is shown in Supplementary Fig. 10b. Before the weaving process, conductive fibres were aligned where gate, drain, and source electrodes will be located. When one transistor is well-aligned, conductive fibres were attached to Mo electrodes by conductive glue (chemical composition in Supplementary Table 4). The area of dispensed conductive glue was carefully controlled to be smaller than Mo electrodes (minimum: $1\ mm \times 3\ mm$).

After conductive glue is left on a target area, laser soldering (XTec, Hacker Automation) was done to instantly solidify conductive glue. The laser spot diameter size and focal length were 400 μm and 10 cm, respectively. During the laser illumination, a real-time camera equipped with a pyrometer checked the temperature of conductive glue (Supplementary Fig. 10c). Measured temperature as a function of irradiation time was optimised for preventing peeling-off of Mo electrode (Supplementary Fig. 10d). The temperature range over 200 °C damaged both Mo electrode and underneath PEN substrate leading to unfavourable catastrophe. The temperature during the laser annealing was kept at $150 \pm 10\ °C$ for a second.

**F-energy storage device fabrication.** F-supercapacitor were prepared by coating solid-gel electrolyte (polyvinyl alcohol (PVA)-$H_3PO_4$) on carbon fibres reported elsewhere[48–50]. PVA (2 g) was dissolved in DI water (20 ml) at 140 °C followed by drop-wise addition of $H_3PO_4$ (2 ml) into PVA solution. This gel electrolyte was continuously stirred for 2 h before coating it on carbon fibres. The electrolyte was coated on straightened carbon fibre (60 cm-long) by manual brushing, then dried at room temperature for 6 h (3 times). Two carbon-electrolyte fibres were twisted together to form an electrical double layer. To prevent short-circuit or breakdown of capacitive layer, twisted fibres were coated by electrolyte solution twice. To increase the total capacitance of F-energy storage device, we added conductive fibre into carbon fibres bundle, which exhibits a 3-fold capacitance increase (Supplementary Fig. 12). Other coating and twisting remained the same as the procedure in this section.

## Data availability
Data associated with this work are available at: https://doi.org/10.17863/CAM.78457.

## Code availability
The codes used for the smart textile system in this work are available at: https://doi.org/10.17863/CAM.78457.

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

## Acknowledgements

This work was supported by the European Commission (H2020, 1D-NEON, Grant agreement ID:685758). J.M.K. and L.G.O. acknowledge support from the UK Research and Innovation (EPSRC, EP/P027628/1). M.C. acknowledges Leverhulme Trust Grant (RPG-2019-227), UK Research and Innovation grant (EPSRC EP/T001038/1), and Royal Society Wolfson Merit Award. H.W.C. thanks to Dr. In Taek Han for fruitful advice on system integration strategy for presenting final smart textile architecture.

## Author contributions

H.W.C. and J.M.K. conceived the project. H.W.C., G.A., L.G.O., R.Ma., M.Chh., and J.M.K. supervised the project. H.W.C., D.S., J.Y., and S.L. carried out the interconnection, programming/coding, and weaving of fibre devices. C.F., R.B., R.I., and P.B. developed fibre-metal oxide thin film transistor. U.E., S.S., S.N., and A.T. developed planar/woven RF antenna. K.U. developed highly conducting fibres. H.L. developed a fibre photo-detector and optimised it for weaving process. S.J., S.D.H., S.Y.B., S.Z., W.H., Y.S., X.F., T.H.L., M.Cho., Y.C. performed device morphology analysis, light-emitting diode characterization, antenna property measurement, and touch sensor reliability test. P.J., F.M.M., V.G.C. carried out weaving of upgradable textile units. M. S. developed conductive glue for interconnection. D.J., A.M., R.Mo., J.G., N.D., A.N., K.C., S.Cho, C.K., K.S. developed weaving pattern, interconnection method, large scale device fabrication, and whole system integration. M. L. assisted in the design of lighting/display unit integration. Y.L., S.Cha, J.I.S. developed fibre energy storage device. H.W.C., D.S., S.L., J.Y., M.Chh., and J.M.K. wrote the paper. All authors discussed the results and commented on the manuscript.

## Competing interests

The authors declare no competing interests.

## Additional information

[1]Electrical Engineering Division, Department of Engineering, University of Cambridge, Cambridge, UK. [2]i3N/CENIMAT, Department of Materials Science, NOVA School of Science and Technology and CEMOP/UNINOVA, NOVA University Lisbon, Campus de Caparica, Caparica, Portugal. [3]Bioelectronics and Advanced Genomic Engineering (BIOAGE), Lamezia Terme, Italy. [4]Textile Research Institute Thuringia-Vogtland (TITV), Greiz, Germany. [5]Eurecat, Centre Tecnológic de Catalunya, Unitat de Teixits Funcionals, Mataró, Spain. [6]Solvay Specialty Polymers s. p. a, Bollate, Italy. [7]SAATI S.p.A, Appiano Gentile, Italy. [8]Centre for Nanotechnology and Smart Materials (CeNTI), Vila Nova de Famalicão, Portugal. [9]LG Display Co., Ltd, Seoul, South Korea. [10]Lighting Applications, Signify, Eindhoven, Netherlands. [11]Silvaco Europe, St. Ives, UK. [12]Relats S. A, Barcelona, Spain. [13]Henkel AG & Co. KGaA, Düsseldorf, Germany. [14]Department of Engineering Science, University of Oxford, Oxford, UK. [15]Department of Energy Systems, Soonchunhyang University, Asan, South Korea. [16]Department of Physics, Sungkyunkwan University, Suwon, South Korea. [17]Division of Physics and Semiconductor Science, Dongguk University, Seoul, South Korea. [18]Samsung Advanced Institute of Technology, European Open Innovation (SAIT-Europe), Samsung Electronics Co., Ltd, Surrey, UK. [19]Department of Materials Science and Metallurgy, University of Cambridge, Cambridge, UK. [20]These authors contributed equally: Hyung Woo Choi, Dong-Wook Shin, Jiajie Yang, Sanghyo Lee. ✉email: lgo23@cam.ac.uk; mc209@cam.ac.uk; jmk71@cam.ac.uk

