## [Peer Review File · Nature Communications]

Smart textile lighting/display system with multifunctional fibre devices for large scale smart home and IoT applicationsREVIEWER COMMENTS

Reviewer #1 (Remarks to the Author):

This article reports a fully operational textile system consisting of display, power transmission, touch sensing, photodetection, signal monitoring and energy storage. By arranging the conductive yarns in the fabric, the authors creatively present textile system towards revolutionary applications on smart homes and internet of things. This is an impressive work in the field of smart textile, so I recommend the publication on Nature Communications. I would prefer that the authors consider addressing the following issues:

1. Washing durability is important in practical application, however, as shown in Supplementary Fig. 25, display textile partly failed after only three cycles of washing, and no washing result was presented about sensing, photodetection and energy storage textile. I suggest that authors add some comments or experimental results on washing performance.
2. The mechanical stabilities of devices were tested under bending radius ranging from 1 to 20 mm. Some explanations should be offered why different bending radius were selected.
3. As Supplementary Fig. 10 shows, electrical connections were formed based on Ag glue. Some evidences are preferred that Ag glue connections can provide sufficient adhesion under deformation.

Reviewer #2 (Remarks to the Author):

This manuscript reports attractive results in terms of system level integration. It is not very clear about the novelty in the device design and novelty, where there are many discrete devices in the textile system. Secondly, the novelty of the materials and the fabrication technology is also not clearly stated. I have the next few questions and suggestions.

1. In addition to Fig 1a, it is suggested to move some of the real zoom-in photos of each F-device from Supplementary to Fig 1., in order to have a clear idea about the actual status of the integrated system.
2. Please indicate the 7 F-devices on the smart tactile system more clearly in Figure 1c.
3. The current design of system-level integration looks like a simple combination of different fabric-based devices located in separate regions. Can the author comments on the possibility of merging these devices into the same region? What are the strategies to minimize the crosstalk and ensure reliability? What are the other limits?
4. More discussion about the significance of integrating these electronics together into a textile system should be provided in the introduction part.
5. How about the fabrication process of the double twisted fiber supercapacitor? Is there any machine for fabrication? If so, may the authors show the photograph of the fabrication machine?
6. In the stability tests, the bending/stretching cycles generally should be more than a few hundred to demonstrate its long-term working stability.
7. Speaking of textiles, washability is a key property of concern in practical applications. Are those F-devices washable?
8. what is the core competitiveness of this textile display since the six single devices are very common and they are just integrated onto the textile platform?

Dear Reviewers

Manuscript number: NCOMMS-21-28418A

Thank you sincerely for taking the time to evaluate our manuscript entitled “Smart textile lighting/display system with multifunctional fibre devices for large scale smart home and IoT applications”. We truly appreciate the in-depth consideration and invitation to a revision of our work for the publication in Nature Communications that reviewers have given to our manuscript.

We would like to submit this point-by-point response letter to express the novelty of our work and measurement/discussion for better understanding of the device mechanism and generality, which reviewers had raised. In the main article, we have included all changes in the manuscript text file with coloured highlighting.

Revision lists

	Original draft	Page	After revision
65 line, 4 page	wearable textile display	62 line, 4 page	Added: wearable single coloured textile display
67 line, 4 page	Reviewer’s request	66-70 line, 4 page	Added: The systematic integration of versatile fibre devices into textile over large scale could be realised that meets harsh requirements including (i) material/device design compatible with textile technology, (ii) non-destructive weaving pattern for fibre device, (iii) the interconnection method applicable textile platform, and iv) instant expression of visual signal from F-device to F-display by signal processing/coding.
71 line, 4 page	six input devices including	72-73 line, 4 page	Added: six input devices, which are compatible with symmetric and asymmetric weaving pattern
Next to 78 line, 4 page	Reviewer’s request	80-85 line, 5 page	Added: F-LED, F-energy storage, F-temperature were woven at one time while F-RF antenna, F-biosensor module, F-photodetector, F-touch sensor were integrated as Lego-like manner. This Lego-like design is to suggest, (i) post-upgradability, (ii) expanding smart textile system to hundred-inch wide, (iii) seamless operation of textile display with additional textile gadgets. As we have realised our textile system for smart home applications, our prototype of

			46-inch system is expected to be much larger when used for real-life applications.
158 line, 7 page	ON/OFF	165 line, 8 page	Revised: on/off
181 line, 8 page	cyclic bending	188 line, 9 page	Added: cyclic bending under 10 mm radius
183 line, 8 page	characteristics over a year	190 line, 9 page	Added: characteristics over a year (Supplementary Fig. 13)
185 line, 8 page	(Supplementary Fig. 13)	192 line, 9 page	Revised: (Fig. 3a)
187 line, 8 page	2.32×10^6 S/m) against less than 1 mm bending	193 line, 9 page	Revised: 2.32×10^6 S/m, Fig. 3b) against 10 mm bending
187 line, 8 page	performed (Fig. 3a)	194 line, 9 page	Revised: performed (Supplementary Fig. 14)
196 line, 9 page	Fig 3b). At the bias voltage of 10 V during 250 bending cycles, off-current was slightly increased, compared to Fig. 2b. On-current was continuously appearing near 100 nA (relative standard deviation, 3.65%) that confirmed the electrical stability.	201-204 line, 9 page	Revised: Fig. 3c). At the bias voltage of 10 V, off-current and on-current were continuously maintaining near 9.3 nA (relative standard deviation: 5.82%) and 278 nA (relative standard deviation: 8.62%), respectively that confirmed that the electrical stability during 1000 bending cycles.
202 line, 9 page	(Fig. 3c), 3000 touch attempts	208-209 line, 9 page	Revised: (Fig. 3d), 1000 touch attempts (3000 touch results in Supplementary Fig. 7h)
205 line, 9 page	thousands of touches	211 line, 9 page	Added: thousands of repeating touches
Next to 205 line, 9 page	F-RF mechanical test was not included.	212-216 line, 10 page	Added: F-RF antenna is made of conductive fibres, which shows slight resistance change during 1000 bending cycle (Fig. 3e). As comparable with single conductive fibre characteristics in Fig. 3b, the electrical current along with spiral square antenna was maintained that indicated the unchanged antenna property in term of output voltage, output power, rectified signal against the distance (Supplementary Fig. 16).
206 line, 9 page	conditions of less than 1mm radius (Fig. 3d)	217 line, 10 page	Revised: conditions of 10 mm radius (Fig. 3f)

215 line, 9 page	leakage current	226 line, 10 page	Added: leakage current under mechanical stress
215-217 line, 9 page	To identify the working condition limit, several bending conditions were tested resulting in 7 mm bending radius preserving initial transfer characteristics that is suitable for F-biosensor module (Fig. 3e)	227-229 line, 10 page	Revised: To identify the working condition limit, 1000 cycle bending with radius of 10 mm (Fig. 3g, lower bending conditions in Supplementary Fig. 11e) was applied resulting in unnoticeable transfer characteristics change ($\Delta V_{TH} = -0.92$ V, on/off= 10^7) that is suitable for F-biosensor module.
221 line, 10 page	damage-free from bending radius of more than 1 mm	233-234 line, 10 page	Revised: damage-free from 1000 bending cycles (radius of 10 mm)
222 line, 10 page	viable (Fig. 3f).	235 line, 10 page	Revised: viable (Fig. 3h).
226 line, 10 page	ideal capacitance retention.	238 line, 10 page	Added: ideal capacitance retention (Supplementary Fig. 12)
228 line, 10 page	test for three functional textiles (Supplementary Fig. 16);	240-241 line, 11 page	Revised: test for three functional textiles which are subjected to be touched, pressed or possible fatigue over a time without protective layer (Supplementary Fig. 17)
252 line, 11 page	Fig. 3b, Supplementary Fig. 17	265 line, 12 page	Revised: Fig. 4b, Supplementary Fig. 16
282 line, 12 page	Supplementary Fig. 12b	295 line, 13 page	Revised: Supplementary Fig. 12a
Next to 286 line, 12 page	Reviewer's request	299-302 line, 13 page	Added: The standard water resistance property of all fibre devices in our textile system have been investigated under IPX7 condition (1 m deep, 30 min) and revealed that all F-devices exhibited unnoticeable performance change (Supplementary Movie 9, Supplementary Fig. 25).
604 line, 26 page	a , F-devices.	620-621 line, 26 page	Revised: a , Schematics of F-devices and corresponding fabricated F-devices.
617-623 line, 28 page	a , Cyclic stretching test (20% strain) of conductive fibres. b , Cyclic bending test ($r_{\text{bending}} = 10$ mm) of F-photodetector showing clear on/off classification. c , 3000 touch attempts for evaluating surface weariness of F-touch sensor. d , Bending test of F-	634-640 line, 28 page	Revised: 1000 cyclic bending test with 10 mm radius. a , F-LED showing no current density change. b , Stable current of conductive fibre. c , F-photodetector showing clear on/off classification. d , Cyclic touch attempts for evaluating surface

	temperature sensor showing stable electrical performance. e, Bending limit of flexible fibre transistor used for F-biosensor module (safe zone > 5 mm). f, The constant capacitance of fibre supercapacitor under bending radius more than 1 mm and electrochemical stability of F-energy storage under 1000 cyclic charge-discharge test.		weariness of F-touch sensor. e, Stable resistance value of RF-antenna (Ag-PA conductive fibre). f, F-temperature sensor showing stable classification of measured temperature range as a function of electrical current. g, Unnoticeable transfer characteristics change of fibre transistor used for F-biosensor module. h, Mechanical damage-free from the bending for fibre supercapacitor.
--	--	--	---

Pont-by-point response

- **Reviewer 1:** This article reports a fully operational textile system consisting of display, power transmission, touch sensing, photodetection, signal monitoring and energy storage. By arranging the conductive yarns in the fabric, the authors creatively present the textile system towards revolutionary applications on smart homes and the internet of things. This is an impressive work in the field of smart textile, so I recommend the publication on Nature Communications.

Response:

We are grateful to Reviewer 1 for these positive remarks, and we hope that he/she finds the revised manuscript as insightful and important to the scientific community as the original draft.

Comment 1: Washing durability is important in practical application, however, as shown in Supplementary Fig. 25, display textile partly failed after only three cycles of washing, and no washing result was presented about sensing, photodetection and energy storage textile. I suggest that the authors add some comments or experimental results on washing performance.

Response:

Reviewer 1's comment on washability was fruitful for us to build the water-resistance standard of our smart textile system. Even we have claimed that **“we pre-defined that our textile system is not for the wearable application but mainly aiming for indoor smart home application, washing is not critical. We suggest that our textile system can be dry-cleaned, or air-cleaned (Samsung Air Dresser or LG styler) instead of water-washing”** in Supplementary information (page 5), we further performed the experiment on water-resistance test based upon international electrotechnical commission (IEC) 60529, which demonstrates the degree of protection against water (IPX rating code) of a given device or gadget. Similar to standard IP indication for mobile phones, this test clearly shows the water-resistance level of our smart textile system.

We characterised the standard water resistance property of all fibre devices woven in our smart textile system which procedure is included in Supplementary Video 9 (1 metre deep, 30 minutes) that satisfies IPX7 rating. The characteristics of fibre devices including conductive fibre, F-LED, F-RF antenna, F-photodetector, F-Touch sensor, F-temperature sensor, F-transistor, and F-energy storage have not been changed after 1-metre water immersion for 30 min. In the revised manuscript, we have added the following discussions **(line 15-18, page 12 in revised manuscript)**; **“The standard water resistance property of all fibre devices in our textile system have been investigated under IPX7 condition (1 m deep, 30 min) and revealed that all F-devices exhibited unnoticeable performance change (Supplementary Movie 9, Supplementary Fig. 25)”** and **supplementary text (line 39-45, page**

5); “To further explore practical water-resistance of F-devices, we investigated water immersion test under IPX7 condition (1 m deep water, 30 min) based on IEC60529 that reveals the degree of protection against water. After leaving F-LED, F-RF antenna, F-photodetector, F-touch sensor, F-temperature sensor, F-transistor, and F-energy storage devices for 30 min, all F-devices shows unnoticeable performance change including conductivity, current, photocurrent, transfer characteristics, impedance, and total capacitance that satisfy IPX7 requirements (Supplementary Video 9, Supplementary Fig. 25).” We understand that further studies are required for machine-washing compatible device design and we plan to report it in near future.

Fig. S25. Water resistance test (IPX7, 1m deep, 30 min) for all fibre-based devices integrated into smart textile system.

Comment 2: The mechanical stabilities of devices were tested under bending radius ranging from 1 to 20 mm. Some explanations should be offered why different bending radius was selected.

Response: Reviewer 1’s comment is very insightful. In accordance with Reviewer 1’s advice, we have now unified bending radius to 10 mm for all devices in the smart textile system and re-measured bending characteristics in Fig. 3. We have also included F-RF antenna bending test which is a conductive fibre-based device to clarify electrical characteristics under bending conditions. We have added sentences in the revised manuscript; “At the bias voltage of 10 V, off-current and on-current were continuously maintaining near 9.3 nA (relative standard deviation: 5.82%) and 278 nA (relative standard deviation: 8.62%), respectively that confirmed the electrical stability during 1000 bending cycles” (line 14-16, page 9), “F-RF antenna is made of conductive fibres, which shows slight resistance change during 1000 bending cycle (Fig. 3e). As comparable with single conductive fibre characteristics in Fig. 3b, the electrical current along with spiral square antenna was maintained that indicated the unchanged antenna property in terms of output voltage, output power, rectified signal against the distance (Supplementary Fig. 16)” (line 24-25, page 9), and “1000 cycle bending with a radius of 10 mm (Fig. 3g, lower bending conditions in Supplementary Fig. 11e) was applied resulting in unnoticeable transfer characteristics change ($\Delta V_{TH} = -0.92$ V, on/off = 10^7)” (line 13-15, page 10). During the revision process, mechanical stability results in Supplementary Figs. 7,11,13 have been changed. Specifically, a bending radius lower than 10 mm (fibre transistor of F-biosensor) was moved to Supplementary Fig. 11f for characterising the bending limit. The large number of touch attempts (3000 touches) revealing abrasion durability of F-touch sensor is moved to Supplementary Fig. 7h.

Fig 3. Mechanical and electrical stability of fibre devices.

Comment 3: As Supplementary Fig. 10 shows, electrical connections were formed based on Ag glue. Some evidences are preferred that Ag glue connections can provide sufficient adhesion under deformation.

Response: We are grateful for Reviewer 1's insightful comments and suggestions. The adhesion strength was one of the critical parameters for the integration of F-devices into a single textile system, which aims to exhibit foldable/rollable manipulation. We had investigated the uniaxial stress test between a conductive fibre and laser-soldered Ag glue. The result shows that a conductive fibre holds up to 350 kPa shear stress which is sufficient to maintain stable signal delivery during the mechanical deformation. In accordance with Reviewer 1's suggestion, this result is added to Supplementary Fig. 10e,f.

Supplementary Fig. 10. A schematic illustration of F-biosensor module and its interconnection within cotton textile with conductive threads. **Added: e, Strength of laser-soldered Ag glue holding a conductive fibre. f, Photos of Ag glue on drain electrode of IGZO transistor before/after shear stress test.** in line 8-9, page 15 (revised Supplementary Fig. 10).

- **Reviewer 2:** This manuscript reports attractive results in terms of system level integration. It is not very clear about the novelty in the device design and novelty, where there are many discrete devices in the textile system. Secondly, the novelty of the materials and the fabrication technology is also not clearly stated. I have the next few questions and suggestions.

Response: Reviewer 2's overview comment is equivalent to comment 8. We will be grateful if Reviewer 2 read our response to comment 8 where we critically evaluate the novelties in our study including materials, device fabrication, system integration technology.

Comment 1: In addition to Fig 1a, it is suggested to move some of the real zoom-in photos of each F-device from Supplementary to Fig 1., in order to have a clear idea about the actual status of the integrated system

Response: We are grateful for Reviewer 2's suggestion. In accordance with Reviewer 2's comment, we have moved real zoom-in photos of each F-device from Fig. 4 to Fig 1.

Comment 2: Please indicate the 7 F-devices on the smart tactile system more clearly in Figure 1c.

Response: We thank Reviewer 2's comment. In accordance with Reviewer 2's suggestion, we have modified Fig. 1c.

Comment 3: The current design of system-level integration looks like a simple combination of different fabric-based devices located in separate regions. Can the author comments on the possibility of merging these devices into the same region? What are the strategies to minimize the crosstalk and ensure reliability? What are the other limits?

Response: We are grateful for Reviewer 2's comment where Reviewer 2's comment is partially correct that the current design of system-level integration is a partial combination of different fabric-based devices located in a separate region. In order to make our smart textile system aiming for being a cornerstone of future smart textile, we designed system architecture of (i) input device block including temperature, touch, biosignal, photonic sensors, (ii) controlling device block (RF and

energy storage) and (iii) output device block with RGB LED textile lighting/display for the future smart IoT.

In detail, F-LED, F-energy storage, F-temperature were woven at one time while F-biosensor module, F-RF, F-photodetector, F-touch sensor were integrated as Lego-like manner. This Lego-like design was to suggest, (i) post-upgradability, (ii) expanding smart textile system to hundred-inch wide, (iii) seamless operation of textile display with additional textile gadgets. As we aimed to utilise our textile system for smart home IoT applications, this prototype of 46-inch system is expected to be much larger when used for real-life applications.

Of course, Reviewer 2's comment is correct that we can merge all of F-devices into the same region during the one-time continuous weaving process. In this study, we have worked with industrial associates as you see the affiliations in the author list. The maximum width of the weaving machine is 3 metre-wide. In order to suggest a textile wall-display (300 inches or larger for digital signage), interconnection between even LED textiles is inevitable. This is the main reason for showing Lego-like integration strategy for three F-devices in our work. We wish that Reviewer 2 sees the design concept of our smart textile system positive as there is no state-of-the-art study accomplishing smart textile system integrated with six F-devices exhibiting wireless power transmission, environment monitoring, biosensor, and energy storage with RGBW colour display in one unit system.

Reviewer 2's comments on crosstalk and reliability are very insightful. These two parameters have been of great concern during the system-level architecture setup. However, the crosstalk has not been found in any F-devices because the inductive/capacitive coupling did not occur. The inductive F-RF antenna operation scheme was not influenced by other F-devices because they are based on DC-current flow through highly conductive fibres (ELITEX, in-house), which does not overlap on RF-antenna by pre-defined weaving pattern. Longer distance (a few metres or higher) remote wireless operations for future IoT application, frequency-dependent (directionality) crosstalk should be carefully considered in near future. We understand that this crosstalk will be common to other technology such as flexible, printable, organic electronics at the far-field power transmission. We will consider this topic as a future study in the smart textile field.

In terms of reliability, we have shown the 1-year material/device stability of F-LED, Ag-Pa fibre, IGZO, ZnO/graphene, Ta₂O₅/SiO₂ dielectric, and encapsulation agents (epoxy, CYTOP). Our smart textile system has been invited to Scientific events, display company research meetings, and broadcasting sectors since 2018 until today under an embargo, we are confident that our smart textile system is reliable. In addition, we have added data on IPX7 (see Supplementary Fig. 25) water resistance test for confirming reliability. In accordance with Review 2's comment, we will continue the study with respect to the various reliability standards.

Comment 4: More discussion about the significance of integrating these electronics together into a textile system should be provided in the introduction part.

Response: Reviewer 2's comment is fruitful. In accordance with Reviewer 2's comment, we have added more discussion on the significance of seamless integration of F-devices into a smart textile system in the introduction part. In addition, the method for interconnecting several textiles into smart textile system have been suggested and added in the introduction part as follows: "The systematic integration of versatile fibre devices into textile over large scale could be realised that meets harsh requirements including (i) material/device design compatible with textile technology, (ii) non-destructive weaving pattern for fibre device, (iii) the interconnection method applicable to textile platform, and iv) instant expression of visual signal from F-device to F-display with signal processing/coding." (line 14-18, page 4 in the revised manuscript): 2) "F-LED, F-energy storage, F-temperature were woven at one time while F-RF antenna, F-biosensor module, F-photodetector, F-touch sensor were integrated as Lego-like manner. This Lego-like design is to suggest, (i) post-upgradability, (ii) expanding smart textile system to hundred-inch wide, (iii) seamless operation of

textile display with additional textile gadgets. As we have realised our textile system for smart home applications, our prototype of 46-inch system is expected to be much larger when used for real-life applications.” (line 1-6, page 5 in revised manuscript).

Comment 5: How about the fabrication process of the double twisted fiber supercapacitor? Is there any machine for fabrication? If so, may the authors show the photograph of the fabrication machine?

Response: F-supercapacitors have been fabricated by customised hands-on tools instead of using a machine as described in the method (F-energy storage device fabrication section). Machine-based fabrication/integration processes are under investigation for future article publication. We have clarified the fabrication method as follows, “**The electrolyte was coated on straightened carbon fibre (60 cm-long) by manual brushing (3 times), then dried at room temperature**” (line 8, page 19). For Reviewer 2’s information, we have adapted well-known method (Reference 48, 50) but have firstly expanded its scope to a large scale (length: 60 cm (can be over 100 cm), number of in-house F-supercapacitors: 500 EAs). Moreover, the investigation of the effect on conductive fibre embedded into anode/cathode electrodes to exhibit 3-fold capacitance increase is one of the novelties in F-energy storage devices.

Comment 6: In the stability tests, the bending/stretching cycles generally should be more than a few hundred to demonstrate its long-term working stability.

Response: Reviewer 2’s comment 6 is in line with Reviewer 1’s comment 1 about mechanical stability test. We have re-characterised all F-devices to satisfy both Reviewers requests. The bending radius was unified to 10 mm (Fig. 3) and the bending cycle was fixed to 1000 cycles. We have received material stability comments from other Reviewers in addition to mechanical stability. To validate the long-term stability of F-devices, specifically conductive threads, metal oxide material (IGZO, ZnO), graphene, supercapacitor’s electrolyte, we characterised mechanical test of all F-devices in re-drawn-Fig.3, which are all 2-year-old samples. We emphasise that both material, device, and operational stability were achieved in our study.

Comment 7: Speaking of textiles, washability is a key property of concern in practical applications. Are those F-devices washable?

Response: Reviewer 2’s comment on washability helped us to build a new water-resistance standard of our smart textile system as this comment is in line with Reviewer 1’s comment 2.

We firstly defined that our textile system is not for wearables but for large-area applications in smart homes and IoT applications. In this perspective, similar to curtains at any home, we do not see the water-washing is essential for large textile which does not fit to the washing machine. Moreover, we forecast non-water-washing trend in the near future as there are emerging innovative consumer electronics including Samsung Air Dresser or LG styler (Supplementary page 5, line 37-40). Therefore, we followed international standards in water resistance measures based on IEC60529 (IPX7, 1 m deep water, 30 min) which is a reliable method in the electronic industry (mobile phones, computers, medium-to-large gadgets). All F-devices immersed in the water for IPX7 test show non-noticeable performance change as shown in a new Supplementary Fig. 25, Supplementary Video 9) as discussion follows, “**To further explore practical water resistance of F-devices, we investigated water immersion test under IPX7 condition (1 m deep water, 30 min) based on IEC60529 that reveals the degree of protection against water. After leaving F-LED, F-RF antenna, F-photodetector, F-touch sensor, F-temperature sensor, F-transistor, and F-energy storage devices for 30 min, all F-devices shows unnoticeable performance change including conductivity, current, photocurrent, transfer characteristics, impedance, and total capacitance that satisfy IPX7 requirements (Supplementary Video 9, Supplementary Fig. 25.**

Comment 8: what is the core competitiveness of this textile display since the six single devices are very common and they are just integrated onto the textile platform?

Response: We would like to first emphasise that our textile system shows a novel architecture of systematic integration of six input devices and one output device, which has not been reported anywhere. Especially, full-colour RGB lighting/display within a textile platform is the first report. From material development to system design under interdisciplinary study, we have developed F-device compatible with the conventional textile weaving process. We have summarised the critical requirements of F-devices that should be considered before integration into a textile platform. Reviewer 2's comment of 'just integration' requires lots of additional parameters, compared to flexible technology, during the smart textile fabrication.

We would like to also emphasise that our smart textile system does not have F-devices 'on' a textile platform. Unlike mounting devices 'onto' a textile, our system is formed by weaving/knitting/embroidery without bonding glues that allow the integration of F-devices 'into' a textile. In this perspective, Reviewer 2 may have a different idea on our integration strategy in this study. In order to perform a weaving process for the line-by-line manner to make 46-inch system in our study, there were numerous requirements for F-devices before the textile process. To name a few,

F-device	Requirements that should be considered for textile electronic system architecture and its integration technology by weaving and interconnection
F-LED	 1) Resistance to interlacing force by weft/warp threads 2) Strong adherence of LEDs on flexible, pre-patterned electrode against hitting force by a reed during the weaving process. 3) High luminance/ High resolution 4) Interconnection toward F-devices via conductive fibres (in-house) 5) Potential for large-scale fabrication
F-RF antenna	 1) Conductive fibres for weaving/knitting/embroidery not by printing on textile substrate 2) Stability of conductive particles during textile process (abrasion resistance) 3) Flexible design rule for receiving targeted RF frequency for the textile system architecture
F-photodetector	 1) Weaving pattern modification for active area protection 2) Proper encapsulation layer against surface scratch during weaving 3) Novel electrode design/interconnection with conductive thread within a textile via laser soldering or conductive glue
F-touch sensor	 1) Resistance to surface abrasion by human touch attempts 2) Grid type sensing point realisation by weaving 3) High signal-to-noise ratio for touch recognition after weaving
F-temperature	 1) Responding only to temperature (resistance mode) 2) Localised temperature detection (small active area) 3) Proper materials for detection range (real-life applicable)
F-biosensor	 1) Resistance to interlacing force on a transistor 2) Processing compatibility for electronic textile architecture 3) Interconnection from three terminals (gate, drain, source) to embedded controller (current reader) without bulky measurement tool ($I_{DS} > 1 \mu A$) 4) Alignment of conductive fibres in both lateral/horizontal directions by weaving
F-energy storage	 1) Mechanically stable electrolyte material during weaving process 2) Scalable processing method for maximising total capacitance (length/output voltage definition) 3) Strategy on serial/parallel connection of F-supercapacitors

As we aimed to build a smart textile system fully compatible with traditional textile process (emphasis on weaving), there is always significant striking force generated by a reed (a tool in weaving loom). This implies that the mechanical requirement is not simply flexibility, but each F-device must be designed/fabricated to overcome this striking force that has never been considered for surface-mounted F-devices. Besides, the interlacing force generated by textile platform thread (cotton in our study) pressing down to each F-device must be considered. Our smart textile system is built after modifying all parameters mentioned in the table above to converge semiconducting technology with conventional textile technology. The methodological fabrication rule of F-devices for overcoming physical controlling parameters during the weaving/interconnection process have been systematically studied and under manuscript preparation.

Second, we have a different opinion from Reviewer 2's claim on the generality of six single devices. In order to make smart textile system aiming for being a cornerstone of future smart, we designed system architecture of input device block including temperature, touch, biosignal, photonic sensors, controlling device block (RF and energy storage) and output device block with RGB LED textile lighting/display for the future smart IoT.

To realise a practical smart textile system, our study reports research-level plan-of-action for **system integration strategy (Abstract: Here we report the realisation of a fully operational 46-inch smart textile lighting/display system consisting of RGB fibrous LEDs coupled with multifunctional fibre devices that are capable of wireless power transmission, touch sensing, photodetection, environmental/biosignal monitoring, and energy storage)**, which is our main claim in this study. The strategy includes, (i) material/device re-design, (ii) non-destructive weaving pattern design for six F-devices, (iii) interconnection method compatible with textile system, and (iv) building a visual signal expression strategy from F-device to F-display. In-depth studies have been included in our work point-by-point to show readers a roadmap to real-life applicable textile platform. In this regard, the table below summarises novelty of each F-device.

F-device	Critical evaluation on novelty of each F-device
F-LED	 1) Highest resolution in F-type LEDs (uniquely designed copper fibre pattern, page 5, line 23-25 with RGB full-colour palette) 2) Brightness over 10,000 cd/m² (state-of-the-art single coloured F-EL device: 150 cd/m²) 3) Low operation voltage of < 3V (state-of-the-art EL device: 300 V/AC) 4) Grey-scale control full colour lighting/display 5) Bendable, flexible rollable with full-colour video movie
F-RF antenna	 1) Made by in-house developed conductive fibre (Ag-PA) 2) 1st showcase of mode-change application (page 11, line 12-17, movie 3) 3) Fully compatible with textile technology (weaving/knitting/embroidery) ($L_{measure} = 2.64 \mu H$) 4) Visual signal expression on F-LED (realisation of rectifier embedded textile system)
F-photodetector	 1) Novel design of weaving pattern for F-photodetector 2) Relatively fast on/off (< 3 s) switching behaviour (supplementary Fig. 6) 3) Suggested novel application of F-photodetector for curtain operation under standardised solar irradiation conditions.
F-touch sensor	 1) 1st demonstration of textile touch sensor for IoT application (supplementary Fig. 22, movie 7) 2) In-house developed conductive fibre to form 30 sensory points (page 6, line 20-25) showing 18 ms responding time 3) Direct visualisation of touch input on F-LED in the system instead of the measured current
F-temperature	 1) F-temperature sensor with Cu₂O material has never been reported. 2) Low fabrication temperature of ~ 300 °C, (state-of-the-art F-temperature

	sensor with Ge₁₇As₂₃Se₁₄Te₄₆: above 600 °C) 3) High temperature sensitivity of 0.1 °C (state-of-the-art: 0.1 °C)
F-biosensor	1) Highest voltage gain (28) of the fibre-type amplifier in the literature 2) 1st realisation of F-biosensor to detect a heartbeat in the textile platform 3) Long-term stability over a year 4) Extremely low power consumption (350 nW) as a peripheral module in the smart textile system (Page 8, line 1-2, supplementary Fig. 11)
F-energy storage	1) 1st showcase for operating a switch for 300 sec controlling the textile display (page 12, line 14-17, movie 8) 2) Novel study on conductive fibre-embedded anode/cathode effect on capacitance increase by 3-fold (Supplementary Fig. 12g and page 8, line 10-12) 3) Real-time monitoring of charging status on F-display by system integration targeting smart home application

To name a few novelties in F-devices, we have summarised key advancements in our study in comparison with state-of-the-art studies. On Reviewer 2's comment of lack of novelty, we find it encouraging for textile research society as there are lots of novel areas in this field competing with flexible electronics. We welcome Reviewer 2's comment and will move forward to find more unique novelties in fibre-based devices for the following studies. Fibre electronic field is still immature as state-of-the-art device performance (published in Nature in 2021) is still infant stage compared to the flat, flexible field. However, our study has shown better/promising results compared to previous studies. We wish that Reviewer 2 finds the novelty in our system integration strategy.

In light of our responses regarding Reviewers' comments and minor technical claims, we hope that Reviewers will be willing to consider our revised manuscript to Nature Communications. As one of the reviewers has indicated "By arranging the conductive yarns in the fabric, the authors creatively present textile system towards revolutionary applications on smart homes and internet of things. This is an impressive work in the field of smart textile, so I recommend the publication on Nature Communications.", we are confident that our ground-breaking work will be a good fit for publication in Nature Communications.

Sincerely,

Prof. Jong Min Kim

Professor of Electrical Engineering in 1944

University of Cambridge

REVIEWERS' COMMENTS

Reviewer #1 (Remarks to the Author):

The authors have addressed my concerns, and it can be accepted now.

Reviewer #2 (Remarks to the Author):

The authors have made good efforts in revising and optimizing the content of this manuscript; hence we suggest acceptance in its current form.

Manuscript number: NCOMMS-21-28418-B

Thank you sincerely for taking the time to evaluate our revised manuscript entitled “Smart textile lighting/display system with multifunctional fibre devices for large scale smart home and IoT applications”. We are happy that both reviewers have suggested to accept our manuscript in its current form.

REVIEWERS' COMMENTS

Reviewer #1 (Remarks to the Author):

The authors have addressed my concerns, and it can be accepted now.

Reviewer #2 (Remarks to the Author):

The authors have made good efforts in revising and optimizing the content of this manuscript; hence we suggest acceptance in its current form.

Sincerely,

Prof. Jong Min Kim

Professor of Electrical Engineering in 1944

University of Cambridge